# Multicenter External Validation of the Liverpool Uveal Melanoma Prognosticator Online: An OOG Collaborative Study

**DOI:** 10.3390/cancers12020477

**Published:** 2020-02-18

**Authors:** Alda Cunha Rola, Azzam Taktak, Antonio Eleuteri, Helen Kalirai, Heinrich Heimann, Rumana Hussain, Laura J. Bonnett, Christopher J. Hill, Matthew Traynor, Martine J. Jager, Marina Marinkovic, Gregorius P.M. Luyten, Mehmet Dogrusöz, Emine Kilic, Annelies de Klein, Kyra Smit, Natasha M van Poppelen, Bertil E. Damato, Armin Afshar, Rudolf F. Guthoff, Björn O. Scheef, Vinodh Kakkassery, Svetlana Saakyan, Alexander Tsygankov, Carlo Mosci, Paolo Ligorio, Silvia Viaggi, Claudia H.D. Le Guin, Norbert Bornfeld, Nikolaos E. Bechrakis, Sarah E. Coupland

**Affiliations:** 1Liverpool Ocular Oncology Research Group, Department of Molecular and Clinical Cancer Medicine, University of Liverpool, Liverpool L7 8TX, UK; antonio.eleuteri@gmail.com (A.E.); hkalirai@liverpool.ac.uk (H.K.); heinrich.heimann@gmail.com (H.H.); rumanahussain@hotmail.com (R.H.); bs0u81b1@student.liverpool.ac.uk (C.J.H.); s.e.coupland@liverpool.ac.uk (S.E.C.); 2Department of Medical Physics and Clinical Engineering, Royal Liverpool and Broadgreen University Hospitals NHS Trust, Liverpool L69 3GA, UK; 3Department of Biostatistics, University of Liverpool, Liverpool L69 3GL, UK; ljbcmshe@liverpool.ac.uk; 4Liverpool Bio-Innovation Hub Biobank, University of Liverpool, Liverpool L7 8TX, UK; mtraynor@liverpool.ac.uk; 5Department of Ophthalmology, Leiden University Medical Center, 2333 ZA Leiden, The Netherlands; m.j.jager@lumc.nl (M.J.J.); M.Marinkovic@lumc.nl (M.M.); G.P.M.Luyten@lumc.nl (G.P.M.L.); M.Dogrusoez@lumc.nl (M.D.); 6Rotterdam Ocular Melanoma Study Group, Erasmus University Medical Center, 3008 AE Rotterdam, The Netherlands; e.kilic@erasmusmc.nl (E.K.); a.deklein@erasmusmc.nl (A.d.K.); k.n.smit@erasmusmc.nl (K.S.); n.vanpoppelen@erasmusmc.nl (N.M.v.P.); 7Ocular Oncology, Vitreoretinal Diseases & Surgery, University of California, San Francisco, CA 94143, USA; bertil.damato@gmail.com (B.E.D.); Armin.Afshar@ucsf.edu (A.A.); 8Oxford Eye Hospital, Nuffield Department of Clinical Neurosciences, University of Oxford, John Radcliffe Hospital, Oxford OX3 9DU, UK; 9Department of Ophthalmology, University of Rostock, D-18057 Rostock, Germany; rudolf.guthoff@med.uni-rostock.de (R.F.G.); b_scheef@gmx.de (B.O.S.); vinodh.kakkassery@uni-luebeck.de (V.K.); 10Department of Ophthalmology, University of Lübeck, D- 23538 Lübeck, Germany; 11Ocular Oncology Department, Helmholtz Moscow Research Institute of Eye Diseases, 105062 Moscow, Russia; svsaakyan@yandex.ru (S.S.); alextsygankov1986@yandex.ru (A.T.); 12S.C. Oculistica Oncologica – Ocular Oncology Service, Ente ospedaliero Ospedali Galliera, 16128 Genova, Italy; carlo.mosci@galliera.it (C.M.); paololigorio.82@gmail.com (P.L.); 13DISTAV-Department of Earth, Environment and Life Sciences, University of Genoa, 16132 Genova, Italy; silviaviaggi@gaslini.org; 14Laboratory of Human Genetics, IRCCS Istituto G. Gaslini, 16147 Genova, Italy; 15Department of Ophthalmology, University Hospital Essen, University Duisburg-Essen, 45147 Essen, Germany; Claudia.LeGuin@uk-essen.de (C.H.D.L.G.); Bornfeld@uni-essen.de (N.B.); Nikolaos.Bechrakis@uk-essen.de (N.E.B.); 16Liverpool Clinical Laboratories, Royal Liverpool University Hospital, Liverpool L69 3GA, UK

**Keywords:** eye cancer, uveal melanoma, prognostic model, LUMPO3, discrimination, calibration, C-statistics, survival probabilities, external centers.

## Abstract

Uveal melanoma (UM) is fatal in ~50% of patients as a result of disseminated disease. This study aims to externally validate the Liverpool Uveal Melanoma Prognosticator Online V3 (LUMPO3) to determine its reliability in predicting survival after treatment for choroidal melanoma when utilizing external data from other ocular oncology centers. Anonymized data of 1836 UM patients from seven international ocular oncology centers were analyzed with LUMPO3 to predict the 10-year survival for each patient in each external dataset. The analysts were masked to the patient outcomes. Model predictions were sent to an independent statistician to evaluate LUMPO3’s performance using discrimination and calibration methods. LUMPO3’s ability to discriminate between UM patients who died of metastatic UM and those who were still alive was fair-to-good, with C-statistics ranging from 0.64 to 0.85 at year 1. The pooled estimate for all external centers was 0.72 (95% confidence interval: 0.68 to 0.75). Agreement between observed and predicted survival probabilities was generally good given differences in case mix and survival rates between different centers. Despite the differences between the international cohorts of patients with primary UM, LUMPO3 is a valuable tool for predicting all-cause mortality in this disease when using data from external centers.

## 1. Introduction

Uveal melanoma (UM) is a rare eye cancer occurring in adults, causing liver metastasis in approximately 50% of cases [1]. Patients’ survival is directly related to the presence of hepatic metastases. After detection of metastatic disease, most patients die within a year, with only a few responding to current therapies [2].

There is some evidence that prognostication in UM improves the quality of life of some patients, even when the probability of survival is poor [3,4,5]. Prognostication is an important aspect of patient care, identifying high-risk UM patients requiring special care (e.g., increased frequency of liver surveillance using high-resolution imaging, enrollment in clinical trials of systemic adjuvant therapy including immunotherapies [6]), while allowing low-risk UM patients to be reassured and to have less intensive surveillance. Many predictive factors of metastasis from UM have been identified [3]. Several of these have been incorporated into our prognostic algorithm, the Liverpool Uveal Melanoma Prognosticator Online (LUMPO) (www.lumpo.net) [7]. 

LUMPO was developed to estimate survival probability in patients treated for UM, combining (a) anatomical predictors, such as largest basal diameter of the tumor, tumor thickness, ciliary body involvement and extra-ocular extension; (b) histological predictors, including epithelioid cell type, presence of closed loops and tumor mitotic count; and (c) genetic predictors, including chromosome-3 deletion and polysomy 8q [8,9]. The tool was validated in 2012 [7] with data from a cohort of patients with UM, with a follow up of more than 20 years at the Liverpool Ocular Oncology Clinic (LOOC).

The first externally available version of LUMPO was validated in 2015, at the Department of Ophthalmology, University of Medical Sciences in Poznan, Poland [10]. This validation study concluded that LUMPO is a useful tool for calculating survival probabilities in an individual patient with UM; however, the authors emphasized that the use of cytogenetic data, which were lacking in their analysis, would potentially improve the accuracy of the prognosis. In 2016, LUMPO was externally validated further by examining data from the USA, in a cohort of UM patients treated at the University of California, San Francisco (UCSF) [11]. Evaluation of these data revealed that there were differences between the two cohorts of patients with respect to anatomical and clinical characteristics, probably because these were not defined and measured in the same standardized fashion. There were also differences in the type of treatment provided to UM patients in the two centers, and, furthermore, genetic data were unavailable within the UCSF dataset at that time [11]. Despite these differences, the external validation showed that LUMPO accurately estimated all-cause mortality for UM patients treated at UCSF.

A revised version of LUMPO (called LUMPO3) was created, incorporating not only chromosome 3 but also 8q data and also calculating mortality using competing-risk methodology [12] This aspect is particularly relevant to prognostication in UM subjects, since in frail populations, such as elderly subjects, other causes of death may occur prior to the occurrence of the event of interest, thus preventing its realization. In that study, estimates of crude cumulative incidence from the raw data showed that metastatic death has a different pattern from death from other causes, thereby necessitating the need for a competing-risks model. Such a model facilitates prediction of metastatic death as a distinctive event from other causes of death. LUMPO3 was internally validated using bootstrap resampling [13], a nonparametric method that allows estimation of optimal model performance measures by random sampling with replacement of data used to fit the model. 

The aim of this study was to perform an external validation of LUMPO3 as a tool for estimating all-cause mortality. All-cause mortality was selected as the primary outcome as it is a readily available outcome, obtainable from national records where relevant. All-cause mortality was estimated from LUMPO3 by aggregating the probability of metastatic death and death from other causes. To this end, the Liverpool Ocular Oncology Research Group (LOORG; wwww.loorg.org) facilitated collection of relevant independent data from members of the European Ophthalmic Oncology Group (OOG; www.oogeu.com) and ocular oncology centers located in the USA. 

## 2. Results

### 2.1. Patient Characteristics

The cohort comprised 1836 patients diagnosed with UM (ciliary body and choroidal). These included 1086 patients from Leiden (LUMC), 218 from Rotterdam (EMCH), 138 from San Francisco (UCSF), 138 from Rostock (UHSH), 134 from Moscow (HIED), 73 from Genoa (SCOO), and 49 from Essen (UHE). These data are shown in Table 1 together with characteristics of the original Liverpool dataset that was used for the development of the model for comparison purposes. Pooled estimates across the different cohorts are also provided. For the medians, the method described in [14] has been applied.

As seen in Table 1, compared to patients treated in Liverpool, those treated in Moscow tended to be more frequently female (Binomial Test: *z* = 3.421 (*p* = 0.001)), who were relatively young and with tumors having a greater basal diameter (T Test: *t* = 6.819 (*p* < 0.001) and *t* = 9.017 (*p* < 0.001) respectively). The latter was also true of patients from Genoa (T Test: *t* = 6.885 (*p* < 0.001)). A higher percentage of patients from Leiden (21%) had extraocular melanoma compared to those treated in other centers (Binomial Test: *z* = 52.75 (*p* < 0.001)). The prevalence of UM containing epithelioid cells also differed between the eight groups in which this feature was documented: it was significantly lower in tumors from San Francisco than those in the Liverpool data set (Binomial Test: *z* = 2.147 (*p* = 0.032)), and much lower than those from Rostock (Fisher’s Exact Test (*p* < 0.001)). All UM from Genoa had epithelioid cells present, which is much higher than the Liverpool dataset (Fisher’s Exact Test (*p* < 0.001)). Genetic data for the UM chromosome 3 status were available from all ocular oncology centers with the exception of Rostock (Table 1). Similarly, most centers also provided information concerning the status of chromosome 8q, with the exceptions of Rostock and Essen (Table 1). Of the cohorts with available genetic data, patients from Genoa had a higher percentage of alterations in both chromosome 3 and chromosome 8q than was seen in Liverpool (Binomial Test: *z* = 2.718 (*p* < 0.001) and *z* = 3.45 (*p* = 0.001) respectively). There was a moderate difference between the Liverpool and Rotterdam datasets in the percentage of alterations in chromosome 3 (Binomial Test: *z* = 2.341 (*p* = 0.02)) and significant difference for chromosome 8q (Binomial Test: *z* = 4.46 (*p* < 0.001)). 

The median follow-up period varied between the external cohorts (range, 0.7–5.2 years) with the shortest median follow-up time being from San Francisco (8 months). Kaplan-Meier curves for all-cause mortality based on the Liverpool dataset and the external datasets are shown in Figure 1. The datasets from Essen and San Francisco matched the Liverpool (development) dataset most closely.

### 2.2. Statistical Analyses

#### 2.2.1. Discrimination

The C-statistic, which examines the discriminative capacity of the model, was evaluated for all participating centers yearly up to 4 years (Table 2) [15]. Values ranged from 0.64 (San Francisco) to 0.85 (Essen) at year 1, through to 0.65 (Moscow) to 0.89 (Essen) at year 4. Pooled estimates of discrimination were fairly consistent across the years at 0.72 (0.68 to 0.75) in year 1 and 0.73 (0.70 to 0.77) in years 2 to 4. This indicated generally good ability of the LUMPO3 model to discriminate between patients who died and those who survived, in independent datasets. 

#### 2.2.2. Calibration

Calibration plots showing predicted probabilities of the outcome against actuarial survival estimates are shown in Figure 2. The plots show good agreement between observed and predicted probabilities. Limited event data in the Essen and Genoa datasets account for the wide confidence bands. Data from Leiden suggests that LUMPO3 over-predicted the survival probability while data from Moscow suggests that LUMPO3 under-predicted mortality, although the event rate was relatively low in the Moscow dataset. 

## 3. Discussion

This is the first multicenter, international, collaborative study to validate and demonstrate the value of a multiparameter prognostic tool in UM—i.e., LUMPO3 developed on large well-phenotyped datasets and robust statistical modelling—for the individualized stratification of patients with respect to metastatic risk and all-cause mortality.

To our knowledge, there currently are no other validated, multifaceted tools that take into account clinical characteristics, histopathologic, and genetic data to predict patient prognosis. Such tools are crucial for reliable decision-making for the identification of patients who may possibly be harmed (physically or psychologically) by inappropriate disease management. Although this is not a major concern in cancers that have a relatively good prognosis and have multiple treatment options with proven clinical benefit, it is an important determinant of clinical care.

Numerous prognostic factors have been identified for primary UM. These have been analyzed alone and in combination to predict the risk of metastasis. These factors can be divided into three main categories: clinical, histologic and genetic [16]. The resulting prognostic tools have led to personalized surveillance regimens [3,17,18] and targeted recruitment to clinical trials for adjuvant therapies. 

Prognostic tools that combine multiple factors include the American Joint Committee on Cancer (AJCC) Tumor Node Metastasis (TNM) staging system for UM, which is based on only tumor size, location and extraocular spread. Genetic characteristics of UM are not included in this system as yet [19]. It is possible to improve the accuracy of prognostic tools by multivariable analysis. This is evidenced by the enhanced prognostic accuracy of the AJCC/TNM staging system when chromosome 3 and 8q status are included [20]. A prognostic nomogram combining AJCC/TNM staging, monosomy 3 and 8q gain has been developed but requires further validation using a larger study group [21]. Similarly, the largest basal tumor diameter was shown to provide additional prognostic information independently of the DecisionDx-UM gene expression profile (GEP) tool classification [22]. 

The National Comprehensive Cancer Network (NCCN) Clinical Practice Guidelines in Oncology for UM (National Comprehensive Cancer Network. NCCN Clinical Practice Guidelines in Oncology (NCCN Guidelines®) Uveal Melanoma Version 1.2018. Natl Compr. Cancer Network, Inc. 2018) stratifies patients as having a low, medium or high risk of metastasis based on a combination of anatomic, histologic and genetic features of the primary tumor. However, it would appear that this prognostic method has not been validated as yet. The Predicting Risk of Metastasis in Uveal Melanoma (PRiMeUM) tool employs a multivariate approach to predict the risk of metastasis developing within 48 months of treatment for the primary tumor. An accuracy of ~85% (derived from Area Under the curve of the Receiver Operating Characteristic [AUROC] analysis) was achieved with a logistic regression model using a combination of clinical and genetic factors. However, the PRiMeUM tool also has yet to be externally validated [23]. Further, an artificial neural network has been created to predict survivorship 5 years from brachytherapy. The network incorporates demographic and clinical data only and again used only data collected at a single center. An accuracy of 84% was achieved (c-index 0.81) when 16 neurons were used in the artificial neural network [24]. 

GEP of 12 discriminating genes has been commercialized as DecisionDx-UM (Castle Biosciences) and classifies patients as at low, medium or high risk of metastasis. The GEP tool was validated in prospective multicenter studies [25,26]. The study by Onken et al. examined the correlation between number of events and GEP classification in UM patients with a short follow-up time of 17.4 months (median) [25]. Plasseraud et al., on the other hand, looked at correlations between pathologic characteristics and molecular class in UM patients with a median follow-up of 27.3 months [26]. However, neither of the GEP studies examined for the calibration aspect of providing accurate probability of survival. Despite these limitations, these studies did demonstrate early promise for the role of GEP in decision making in UM. 

These previous experiences attest to the difficulty faced in studying the strongest prognostic factors for UM. The rarity of this disease makes it hard to collect a wide and comprehensive series of the prognostic factors, with great variations being seen in the modalities of diagnosis, histologic and genetic assessments, as well as in treatments during the observational period. However, with the availability of new regional therapies and targeted drugs, a simple and validated model for risk stratification of the patient such as LUMPO3 is urgently needed. 

In this multicenter collaborative study, sufficient data were collected to perform a reliable validation of the prognostic accuracy of the LUMPO3 model. A limitation of this study is the relatively short follow-up time in some centers, because of the rarity of the disease, as well as its retrospective nature. Despite the differences between cohorts, the model’s ability to discriminate between UM survivors and patients who died either from the disease or other causes was fair to good, as was the agreement between observed and predicted survival probabilities in most centers. Therefore, the LUMPO3 model is able to stratify the prognosis for UM patients and appears to be a valuable tool for predicting all-cause mortality in patients with UM. This model may therefore inform physicians’ management when caring for UM patients, allowing for a better allocation of resources with respect to systemic surveillance. 

## 4. Material and Methods

### 4.1. Ethics

This study conformed to the principles of the Declaration of Helsinki. Approval for this study was obtained from the Health Research Authority (NRES REC ref 18/NW/0748) and anonymized data from consented patients were transferred from external centers according to local approvals.

### 4.2. Data Collection

In November 2017, a call for participation in this external validation of LUMPO3 was made to 14 centers involved in OOG and collaborative studies (Figure 3). After an initial expression of interest by 11 centers, seven centers ultimately submitted their data for analysis. The study protocol was shared with the participating centers. The participating centers (Leiden University Medical Centre (LUMC), Leiden and Erasmus Medical Centre Hospital (EMCH), Rotterdam in the Netherlands, University of California San Francisco (UCSF), U.S.A., University Hospital Schleswig-Holstein (UHSH) in Rostock, Germany, the Helmholz Institute of Eye Diseases (HIED) in Moscow, Russia, S.C. Oculistica Oncologica (SCOO) in Genoa, Italy, and University Hospital of Essen (UHE) Germany) were asked to provide the following data: (1) demographic data—sex and age; (2) anatomical data—ultrasound or histopathological measurements of largest basal tumor diameter, tumor thickness, presence or absence of ciliary body involvement and presence or absence of extraocular extension; (3) histological data—presence or absence of extravascular matrix loops, presence or absence of epithelioid cells, and mitotic cell count (MITOC) per 40 high power fields (HPF); and (4) genetic data—chromosome 3 and 8q status. The MITOC was dichotomized as follows: 0—1/40 HPF = 1; 2—3/40 HPF = 2; 4—7/40 HPF = 3; >7/40 HPF = 4. Histological analysis was undertaken by all Centers using standard protocols, as previously described [9]. Full descriptions of how genetic data were obtained and classified (e.g., Fluorescence in situ hybridization (FISH) methods, Multiplex Ligation Probe Amplification (MLPA) [27] or other methods) were also requested. 

Cases were pseudo-anonymized in accordance with local institutional policies and guidelines to export patient data. Cases were excluded if missing data included age, sex or basal tumor diameter as these have been established to be highly predictive of outcome. If any other variables are missing, they can be imputed using a model-projection framework as detailed in Eleuteri et al. 2018 [12]. The result of this imputation process will be reflected upon in the confidence interval—the more missing variables, the wider the interval. The data were transferred to the data manager (co-author MT) at the Liverpool Bio-Innovation Hub (LBIH) Biobank at the University of Liverpool (UoL), where patient identification and outcome were masked before passing the datasets to co-authors ACR and AT for LUMPO3 analysis. Using LUMPO3, ACR and AT predicted outcomes, which were then compared with the actual outcomes by a neutral Biostatistician mediator, LJB, to determine the performance of the LUMPO3 tool (Figure 1). The comparative results were analyzed as below using statistical methods by LJB. 

### 4.3. Statistical Analyses

Characteristics of the Liverpool (development) and external (validation) datasets were visually assessed for agreement. A Kaplan–Meier curve of all-cause mortality was also produced to evaluate event rates across the datasets. 

The LUMPO3 model was designed by co-authors AE and AT to predict the probability of survival at yearly intervals for each UM patient [12]. The survival predictions were sent to the independent statistician (LJB) to undertake external validation using discrimination and calibration methods [28]. Discrimination refers to the ability of the prognostic model to differentiate between patients who died during this study and those who did not. The discriminative capacity of the model was measured using Harrell’s C-statistic [15,29]. It is measured on a scale ranging from 0.5 (no better than chance) to 1 (perfect prognosis). A pooled estimate of discrimination was calculated using a random effects meta-analysis, which accounted for the correlation between studies [29]. Calibration refers to how closely the probability of the event predicted by the model agrees with the observed probability [28]. Calibration was assessed graphically [28]; if predicted and observed probabilities agree over the whole range of probabilities, the plots show a 45° line. Statistical analyses were conducted using R statistical software version 3.5.0. 

## 5. Conclusions

Despite the differences between cohorts, LUMPO3 appears to be a reasonably accurate and valuable tool predicting all-cause mortality in patients with UM. It should be noted that prognostic tools evolve as new information regarding tumor biology accrues. Whilst the genetic information incorporated into LUMPO3 are the copy number variations of chromosome 3 and 8, future versions of our tool are likely to incorporate key mutations as described in primary UM [30]. However, such revisions require sufficient data (and therefore time) for the revised algorithm to be made robust. We are also currently exploring the possibility of recalibrating the model, so that its predictions can be adapted to external data with different baseline hazard rates.

## Figures and Tables

**Figure 1 cancers-12-00477-f001:**
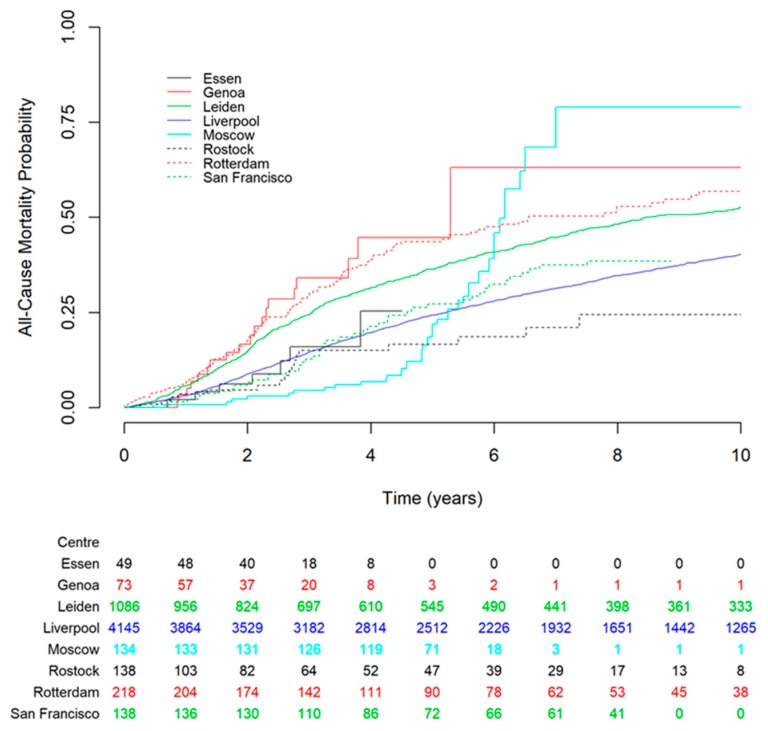
Kaplan–Meier estimates of all-cause mortality for the centers involved in the study. The Liverpool development dataset is shown in solid blue line for comparison. The figure shows that datasets from Essen and San Francisco had the closest match to the Liverpool dataset. The numbers below the figure are the number of subjects at risk entering the corresponding time point for each dataset.

**Figure 2 cancers-12-00477-f002:**
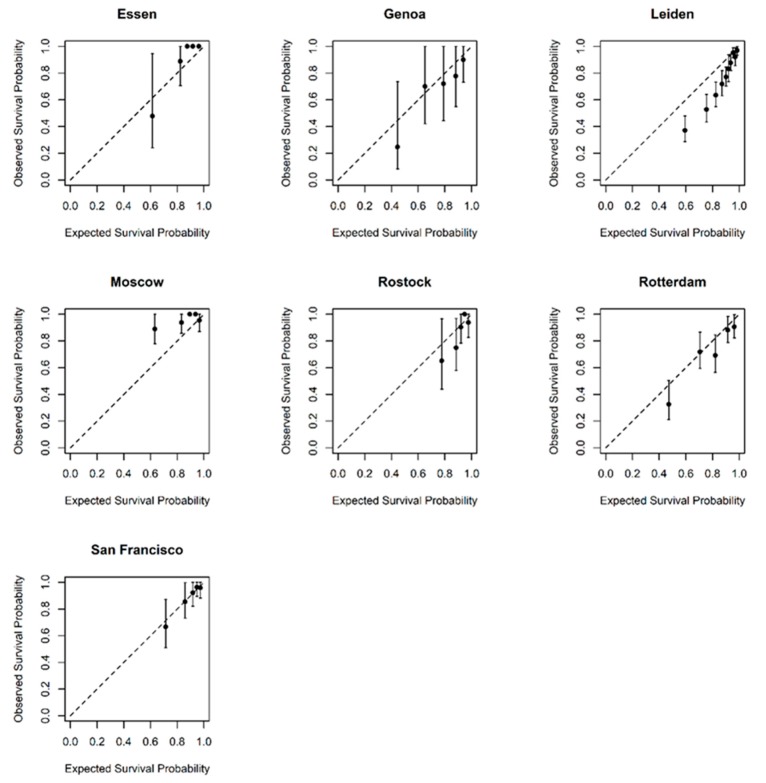
Calibration graphs comparing observed and predicted survival in each of the external datasets at 3 years post-treatment. Subjects were divided into five distinct prognostic groups according to their predicted survival and the average predicted survival for each group was plotted against observed survival. The dashed diagonal line is the line of equality and so markers along this line show perfect agreement between their predicted and observed survival.

**Figure 3 cancers-12-00477-f003:**
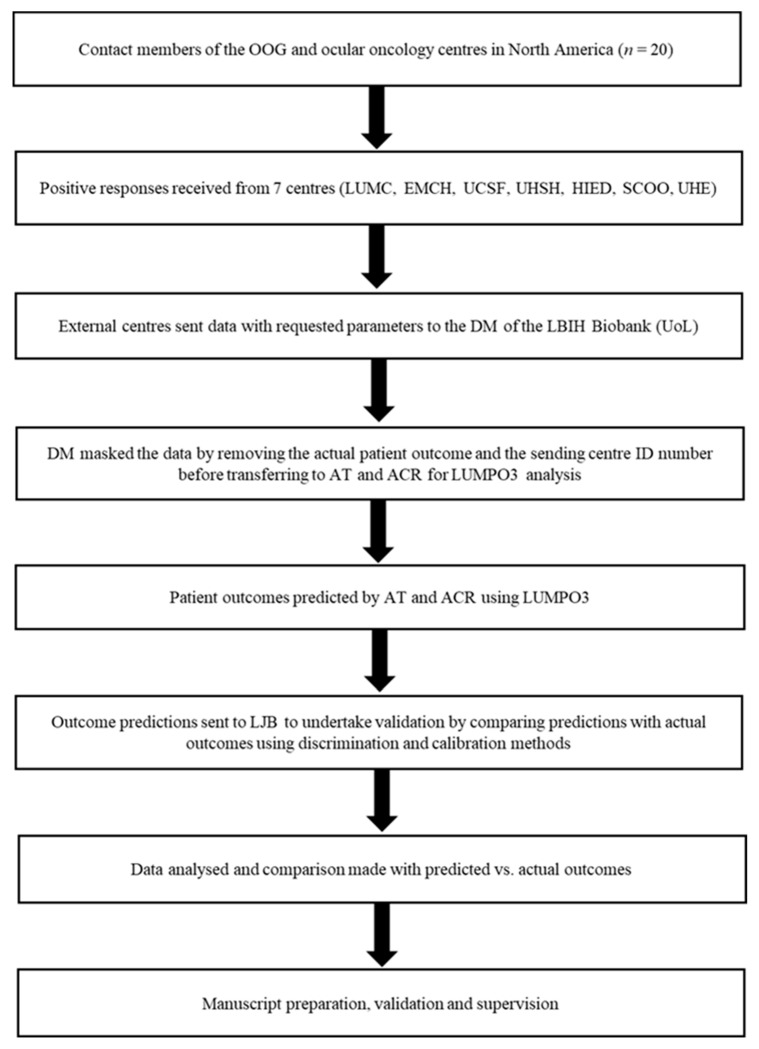
Flow Diagram External Validation LUMPO3. OOG—Ophthalmic Oncology Group, LUMC—Leiden University Medical Centre, EMCH—Erasmus Medical Centre Hospital, UCSF—University of California—San Francisco, UHSH—University Hospital Schleswig-Holstein, HIED—Helmholz Institute of Eye Diseases, SCOO—S.C. Oculistica Oncologica, UHE—University Hospital Essen, DM—Data Manager, LBIH—Liverpool BIO-Innovation Hub Biobank, UoL—University of Liverpool, ID—Identification, AT—Azzam Taktak, ACR—Alda Cunha Rola, LUMPO3—Liverpool Uveal melanoma Prognosticator Online (version 3), and LJB—Laura J. Bonnett.

**Table 1 cancers-12-00477-t001:** Patient characteristics. Development data (Liverpool) and external validation data (from seven ocular oncology centers—Leiden, Rotterdam, San Francisco, Rostock, Moscow, Genoa and Essen). The last column reports the pooled estimates of the characteristics.

Characteristics, *n* (%) Unless Otherwise Stated	Liverpool (*n* = 4145)	Leiden (*n* = 1086)	Rotterdam (*n* = 218)	San Francisco (*n* = 138)	Rostock (*n* = 138)	Moscow (*n* = 134)	Genoa (*n* = 73)	Essen (*n* = 49)	Pooled Estimates
**Age at treatment (years), mean (SD)**	61.4 (14.1)	60.7 (14.4)	62.0 (14.3)	60.0 (13.1)	64.8 (13.8)	53.0 (13.7)	62.0 (16.2)	63.8 (14.7)	61.2 (14.2)
**Sex**									
**Female**	2010 (48)	498 (46)	111 (51)	67 (49)	80 (58)	84 (63)	26 (36)	27 (55)	2903 (48)
**Male**	2135 (52)	588 (54)	107 (49)	71 (51)	58 (42)	50 (37)	47 (64)	22 (45)	3078 (52)
**Missing**	0	0	0	0	0	0	0	0	0
**Largest Ultrasound Diameter (mm), mean (SD)**	12.4 (3.8)	11.2 (3.7)	12.9 (3.6)	11.2 (3.3)	11.3 (3.5)	15.4 (3.3)	15.5 (3.3)	13.8 (3.7)	12.3 (3.7)
**Missing**	110	0	4	4	0	0	0	0	118
**Ultrasound tumour Height (mm), mean (SD)**	5.3 (3.4)	5.6 (3.3)	7.38 (3.5)	5.3 (2.1)	5.2 (2.9)	9.1 (2.9)	10.8 (3.5)	8.6 (3.5)	5.6 (3.3)
**Missing**	98	1	6	0	0	0	1	0	106
**Ciliary Body Involvement**									
**No**	3046 (73)	803 (74)	154 (71)	32 (84)	130 (94)	93 (69)	63 (86)	33 (69)	4354 (74)
**Yes**	1098 (27)	283 (26)	64 (29)	6 (16)	8 (6)	41 (31)	10 (14)	15 (31)	1525 (26)
**Missing**	1	0	0	100	0	0	0	1	102
**Extraocular Melanoma**									
**No**	3872 (93)	848 (79)	191 (88)	134 (99)	130 (96)	119 (89)	73 (100)	35 (92)	5402 (90)
**Yes**	273 (7)	228 (21)	27 (12)	1 (1)	5 (4)	15 (11)	0 (0)	3 (8)	552 (10)
**Missing**	0	10	0	0	3	0	0	11	24
**Epithelioid cells present**									
**No**	915 (42)	351 (33)	74 (34)	38 (55)	31 (97)	61 (46)	0 (0)	-	1470 (39)
**Yes**	1268 (58)	720 (67)	144 (66)	31 (45)	1 (3)	71 (53)	56 (100)	-	2291 (61)
**Missing**	1962	15	0	0	106	2	17	49	2151
**Closed PAS+ Loops**									
**No**	600 (50)	230 (40)	124 (58)	-	-	-	-	-	954 (48)
**Yes**	597 (50)	346 (60)	88 (42)	-	-	-	-	-	1031 (52)
**Missing**	2948	510	0	138	138	134	73	49	3990
**MITOC (n, %)**									
**0**	673 (38)	173 (17)	14 (8)	1 (20)	32 (100)	-	-	-	893 (30)
**1**	414 (23)	282 (28)	27 (16)	0 (0)	0 (0)	-	-	-	723 (24)
**2**	366 (21)	291 (29)	45 (27)	4 (80)	0 (0)	-	-	-	706 (24)
**3**	307 (17)	264 (26)	81 (49)	0 (0)	0 (0)	-	-	-	652 (22)
**Missing**	2385	76	51	133	106	134	73	49	3007
**Chromosome 3 loss**									
**No**	333 (55)	201 (50)	100 (46)	22 (58)	-	77 (57)	27 (39)	37 (76)	797 (53)
**Yes**	269 (45)	202 (50)	117 (54)	16 (42)	-	57 (43)	43 (61)	12 (24)	716 (47)
**Missing**	3543	683	1	100	138	0	3	0	4468
**Chromosome 8 gain**									
**No**	330 (55)	186 (53)	82 (38)	21 (55)	-	97 (72)	23 (34)	-	739 (52)
**Yes**	272 (45)	162 (47)	136 (62)	17 (45)	-	37 (28)	45 (66)	-	669 (48)
**Missing**	3543	738	0	100	138	0	5	49	4573
**Follow-up time (years), median**	6.5	5.2	4.0	0.7	2.7	5.0	2.0	2.7	6.5
**(IQR)**	(3.2–11.7)	(4.3–5.9)	(2.3–8.0)	(0.5–2.1)	(1.0–6.5)	(4.5–5.7)	(1.1–3.0)	(2.1–3.3)	(3.2–11.7)
**Outcome**									
**Alive**	2480 (60)	440 (41)	98 (45)	94 (68)	121 (88)	92 (69)	54 (74)	42 (86)	3421 (57)
**Dead**	1665 (40)	646 (59)	120 (55)	44 (32)	17 (12)	42 (31)	19 (26)	7 (14)	2560 (43)
**Missing**	0	0	0	0	0	0	1	0	1
**Cause of Death**									
**Other**	770 (46)	291 (45)	36 (30)	-	4 (27)	5 (12)	2 (11)	2 (33)	1110 (43)
**Possible UM metastasis**	0 (0)	0 (0)	0 (0)	-	6 (40)	10 (24)	2 (11)	1 (17)	19 (2)
**Definite UM metastasis**	893 (54)	355 (55)	78 (70)	-	5 (33)	27 (64)	16 (84)	3 (50)	1377 (55)
**Missing**	2	0	6	44	2	0	0	1	55

**Legend**: SD = Standard deviation; PAS = Periodic Acid Schiff; MITOC = mitotic cell count (see Methods); IQR = interquartile range; UM = uveal melanoma.

**Table 2 cancers-12-00477-t002:** Discrimination—per year up to 4 years of follow up.

Dataset	1 year	2 year	3 year	4 year
**Essen**	0.85 (0.72, 0.98)	0.87 (0.77, 0.98)	0.89 (0.80, 0.98)	0.89 (0.80, 0.98)
**Genoa**	0.78 (0.68, 0.88)	0.78 (0.68, 0.88)	0.78 (0.69, 0.88)	0.78 (0.69, 0.88)
**Leiden**	0.72 (0.70, 0.74)	0.73 (0.71, 0.75)	0.73 (0.71, 0.75)	0.73 (0.71, 0.75)
**Moscow**	0.65 (0.56, 0.74)	0.64 (0.54, 0.75)	0.65 (0.54, 0.75)	0.65 (0.54, 0.75)
**Rostock**	0.70 (0.57, 0.84)	0.72 (0.59, 0.84)	0.71 (0.57, 0.84)	0.71 (0.58, 0.84)
**Rotterdam**	0.73 (0.69, 0.78)	0.74 (0.69, 0.78)	0.74 (0.69, 0.78)	0.74 (0.69, 0.78)
**San Francisco**	0.64 (0.56, 0.72)	0.66 (0.58, 0.74)	0.66 (0.58, 0.74)	0.66 (0.58, 0.74)
**Pooled estimate**	0.72 (0.68, 0.75)	0.73 (0.70, 0.77)	0.73 (0.70, 0.77)	0.73 (0.70, 0.77)

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
