# Peer review of "Multicenter External Validation of the Liverpool Uveal Melanoma Prognosticator Online: An OOG Collaborative Study"

_cancers, 2020, doi:10.3390/cancers12020477_

Round 1

Reviewer 1 Report

This is a very well done multicenter and international analysis to validate a prediction tool called LUMPO3 for uveal melanoma. They collect data from 1836 patients with uveal melanoma and applied this prediction tool LUMPO3 to validate whether it can be use to determine prediction of survival. The predictions were sent to an independent statistician for model preditions.

The multi-parameter prognostic tool, LUMPO3, was established to stratify patients with respect to metastatic risk and mortality, combining clinical characteristics, histopathology and genetic testing.

While the validation of the prediction tool LUMPO3 seems consistent along all groups tested from the 8 different locations (Liverpool, Leiden, Rotterdam, San Francisco, Rostock, Moscow, Genoa and Essen), the time of the following up study of patients is quite variable. While some locations made the following up of patients from 8 moths (San Francisco) up to 6,5 years (Liverpool). Authors consider these variations do not modify the statistical results, to discriminate between patients that died or survived.

Authors should not only proof that the prediction tool is valid but that also helps to make predictions and this is not clearly stated. It will help the validation of the tool if they show that using the LUMPO3 prediction, in uveal patients they could either decide an specific treatment or another. Authors have not offered any results in this study, only survival or death.

Decide only by a prediction tool concerning survival or death, if you are going to give treatment to those patients with better survival prediction, raises some concerns regarding ethics.

Authors should provide more arguments and data to show the benefits to use the LUMPO3 predictor tool for treatment selection, rather than a survival/death decision. Moreover, authors should provide a Figure where they compile the predictor paramenters, the results achieve and the predictions this tool can provide to clinical physicians in oncology, regarding survival/death but also uveal melanoma treatment.

Therefore, to make LUMPO3 as a tool for uveal melanoma patients treatment, authors should apply this predictor for different treatments 

Author Response

This is a very well done multicenter and international analysis to validate a prediction tool called LUMPO3 for uveal melanoma. They collect data from 1836 patients with uveal melanoma and applied this prediction tool LUMPO3 to validate whether it can be use to determine prediction of survival. The predictions were sent to an independent statistician for model preditions.

The multi-parameter prognostic tool, LUMPO3, was established to stratify patients with respect to metastatic risk and mortality, combining clinical characteristics, histopathology and genetic testing.

While the validation of the prediction tool LUMPO3 seems consistent along all groups tested from the 8 different locations (Liverpool, Leiden, Rotterdam, San Francisco, Rostock, Moscow, Genoa and Essen), the time of the following up study of patients is quite variable. While some locations made the following up of patients from 8 moths (San Francisco) up to 6,5 years (Liverpool). Authors consider these variations do not modify the statistical results, to discriminate between patients that died or survived.

Response: Thank you very much for your comments above. We agree that there are differences in the length of the follow-up; however, this weakness is one associated with such a large multi-centre retrospective study in such a rare tumour. We are of the opinion that excluding patients as suggested by the reviewer would cause bias.

Authors should not only proof that the prediction tool is valid but that also helps to make predictions and this is not clearly stated. It will help the validation of the tool if they show that using the LUMPO3 prediction, in uveal patients they could either decide an specific treatment or another. Authors have not offered any results in this study, only survival or death.

Response: The study proposed by the reviewer would be a whole new study in itself, and is ‘out of scope’ of the current project. We actually suspect that some ocular oncologists estimate prognosis intuitively, without referring to any recognised prognostication systems, e.g. either the TNM/AJCC staging system or LUMPO. The study proposed by the reviewer would have to be prospective in design, and could, therefore, compare such ‘fuzzy logic’ or ‘gut predictions’ with LUMPO and/or TNM predictions.

As mentioned below in more detail, for several years, the Liverpool Ocular Oncology Clinic has been routinely using LUMPO to stratify UM patients into metastatic risk groups and thereby vary their liver surveillance management: the high-risk UM patients receive regular 6-monthly MRI scans whilst the low-risk UM patients underwent less regular ultrasound scans of the liver. Please see also the already cited references 13 and 14.

Decide only by a prediction tool concerning survival or death, if you are going to give treatment to those patients with better survival prediction, raises some concerns regarding ethics.

Response: With respect to the reviewer, we are not sure we understand this comment. We do not restrict ocular treatment to those patients who have a better prognosis only. All patients are provided some sort of medical and mental health support throughout the pathway.

Authors should provide more arguments and data to show the benefits to use the LUMPO3 predictor tool for treatment selection, rather than a survival/death decision. Moreover, authors should provide a Figure where they compile the predictor paramenters, the results achieve and the predictions this tool can provide to clinical physicians in oncology, regarding survival/death but also uveal melanoma treatment.

Response: In 2013, our group published a study (reference 13) in which we could demonstrate that by using the predecessor of our prognosticator tool, UM patients could be prospectively stratified into low- and high metastatic risk groups, with the high-risk patients undergoing a six-monthly assessments, comprising history-taking, clinical examination, hepatic MRI (without contrast, unless suspicious lesions were identified) and biochemical liver function tests. Ninety (48%) of the 188 patients developed detectable metastases, a median of 18 months after ocular treatment. Six-monthly MRIs detected metastases before symptoms in 83 (92%) of 90 patients developing systemic disease, with 49% of these having less than five hepatic lesions all measuring less than 2 cm in diameter. Of these 90 patients, 12 (14%) underwent hepatic resection, all surviving for at least a year afterwards.

This study demonstrated that our multiparameter prognosticator allows for the identification of high-risk UM patients with high accuracy, allowing for six-monthly MRI surveillance and the earlier detection of metastases. In turn, this enhances opportunities for early treatment of metastatic disease (i.e. surgery) and clinical trial participation (see reference 14).

Indeed, with respect to the latter, the Liverpool UM clinical network has led three national clinical trials in the UK addressing metastatic UM (termed ITEM, SUAVE and SelPac). The results of the first two trials have been published as abstracts in conferences (and could not be included as formal publications in the citation list); the third trial is still underway and hence the results are not publicly available as yet.

Finally, survey-based research with UM patients by ourselves and others would suggest that most UM patients feel empowered with the prognostic information, even if their prognosis is poor (PMID: 21029286 and PMID: 19421848). We have added this into the Discussion and included this reference.

Therefore, to make LUMPO3 as a tool for uveal melanoma patients treatment, authors should apply this predictor for different treatments.

Response: With respect, we disagree with the reviewer here. It has been demonstrated quite robustly by several groups in the literature that UM survival is not influenced by type of treatment, and hence there would be no logic in applying the LUMPO tool to the differing treatment groups. Although local recurrence has been associated with an increased risk of metastasis, these cases are quite rare and it would be very difficult to collate sufficient numbers even using such an international collaborative platform.

Reviewer 2 Report

It was with great interest I read the manuscript concerning prognostication in uveal melanoma. The manuscript is clearly written, has a clear aim and conclusion and adds knowledge to the field of uveal melanoma. I am not myself an expert in prognostic models, but my only comment concerns the C-statistics used. Usually for tests like this using binary outcomes in a logistic regression model, I am used to see the ROC-curves and my question is if this can be added to better describe the model? Apart from this, I do not have any additional comments.

Author Response

Response:

Thank you very much for your comments.

With respect to the request for a ROC curve: The C-statistic used in our study is the analogue of the Area under the ROC for time-to-event data (i.e. in the survival analysis setting), and has the same interpretation (please see Taktak et al., 2006 - PMID: 17184760 – added into the Methods).External validation studies like the current one should report both discrimination (using C-statistic or similar) and calibration (calibration plot or similar).  We have provided both in the current study, and so we are of the opinion that we have suitably demonstrated external validity of the model from a statistical point of view. We hope that the reviewer is in agreement with this.